# Dysfunction of Chloroplast Protease Activity Mitigates *pgr5* Phenotype in the Green Algae *Chlamydomonas reinhardtii*

**DOI:** 10.3390/plants13050606

**Published:** 2024-02-23

**Authors:** Shin-Ichiro Ozawa, Guoxian Zhang, Wataru Sakamoto

**Affiliations:** 1Institute of Plant Science and Resources, Okayama University, Kurashiki 710-0046, Japan; ozwsh1r@okayama-u.ac.jp; 2College of Land and Environment, Shenyang Agricultural University, Shenyang 110866, China; 2015500066@syau.edu.cn

**Keywords:** photoinhibition, chloroplast protease, Photosystem I, *Chlamydomonas reinhardtii*

## Abstract

Researchers have described protection mechanisms against the photoinhibition of photosystems under strong-light stress. Cyclic Electron Flow (CEF) mitigates electron acceptor-side limitation, and thus contributes to Photosystem I (PSI) protection. Chloroplast protease removes damaged protein to assist with protein turn over, which contributes to the quality control of Photosystem II (PSII). The PGR5 protein is involved in PGR5-dependent CEF. The FTSH protein is a chloroplast protease which effectively degrades the damaged PSII reaction center subunit, D1 protein. To investigate how the PSI photoinhibition phenotype in *pgr5* would be affected by adding the *ftsh* mutation, we generated double-mutant *pgr5ftsh* via crossing, and its phenotype was characterized in the green algae *Chlamydomonas reinhardtii*. The cells underwent high-light incubation as well as low-light incubation after high-light incubation. The time course of Fv/Fm values in *pgr5ftsh* showed the same phenotype with *ftsh1-1*. The amplitude of light-induced P700 photo-oxidation absorbance change was measured. The amplitude was maintained at a low value in the control and *pgr5ftsh* during high-light incubation, but was continuously decreased in *pgr5*. During the low-light incubation after high-light incubation, amplitude was more rapidly recovered in *pgr5ftsh* than *pgr5*. We concluded that the PSI photoinhibition by the *pgr5* mutation is mitigated by an additional *ftsh1-1* mutation, in which plastoquinone pool would be less reduced due to damaged PSII accumulation.

## 1. Introduction

Under light, P680 in Photosystem II (PSII) and P700 in Photosystem I (PSI) are oxidized and a photosynthesis reaction takes place. PSII oxidizes water molecules in the Mn_4_CaO_5_ cluster, and the electrons from the water molecules are transferred via electron transfer pathway [1]. Most of the cofactors are ligated in D1/D2 heterodimer, and reduce plastoquinone to generate plastoquinol in the thylakoid membrane. The Cytochrome *b*_6_*f* complex (*b*_6_*f*) transfers electrons from plastoquinol to plastocyanin, coupled with proton translocation from the stroma to the lumen. The *b*_6_*f* reduces plastoquinone, consuming the stromal proton, and oxidizes plastoquinol, releasing the proton to the lumen [2]. The oxidized P700 (P700^+^) receives an electron from plastocyanin, and the electron transfer reaction within PSI eventually reduces ferredoxin (Fd) in the stroma. The Ferredoxin NADP(+) reductase receives an electron from Fd and synthesizes NADPH. This electron transfer coupled with proton translocation forms a proton gradient (ΔpH) across the thylakoid membrane that is essential for ATP synthesis by chloroplast ATP synthase [3]. ΔpH also induces photoprotection mechanisms such as non-photochemical chlorophyll fluorescence quenching (NPQ) [4]. Throughout this overall electron transfer reaction, linear electron flow (LEF) will generate oxygen, ATP, and NADPH.

Under strong light, both photosystems are potentially photoinhibited and need to be protected [5]. Reactive oxygen species (ROS) are generated under strong-light conditions and damage proteins. PSII is sensitive to high-light stress, and several of its protection mechanisms have been described [6,7]. The main mechanism is NPQ, which includes energy dissipation, the xanthophyll cycle, state transition, and antenna size regulation. In addition, the antioxidant reaction can be considered an important mechanism. Regardless of these protective mechanisms, PSII is prone to damage (even in weak light). Therefore, PSII repair is required to maintain its function [8,9]. The D1/D2 protein heterodimer accommodates cofactors that form an electron transfer pathway within PSII. It is known that damaged D1 protein is degraded by chloroplast protease, and then, replaced with newly synthesized D1 protein to maintain the functional structure of the electron transfer pathway. FTSH is described as a metalloprotease that predominantly degrades damaged D1 protein, and is conserved in plants and algae [10,11,12]. In the green algae *Chlamydomonas reinhardtii*, FTSH1 and FTSH2 localized in chloroplasts and FTSH1/2 form a hetero hexamer that enables them to function. *ftsh1-1* is isolated as a loss-of-function mutant which accumulates FTSH1 protein, carrying an amino acid substitution on the 420th Arginine to Cysteine and impairing the oligomerization of FTSH1/2. *b*_6_*f* is also an FTSH substrate [12]. It is reported that the oxidation of the stromal-side D1 subunit is important for the interaction with FTSH [13] and several oxidation sites are found in PSI (PsaA/B/C/D/E/F/H/J/L) and LHCI subunits (Lhca1/2/3/4) in field-grown spinach [14]. However, while proteases involved in PSI degradation have not yet been described, an in vitro assay reports that metalloprotease could exist [15]. It is reported that ROS induces an overreduction in electron transfer carriers that results in PSI photoinhibition [16,17].

Electron flow regulation contributes to reduce photoinhibition. Photosynthetic control increases the electron-flow rate within *b*_6_*f* when the plastoquinone pool is fully reduced and helps to generate sufficient ΔpH to establish NPQ. A photosynthetic-control deficient-mutant accumulates more P700^+^ under strong light and does not establish NPQ [18,19,20]. The electrons of Fd are reinjected into a plastoquinone pool to form cyclic electron flow (CEF) between *b*_6_*f* and PSI, where the electrons are recycled to protect PSI. CEF contributes to the augmentation of the proton gradient across thylakoid membrane, and as a result, more ATP is synthesized without NADPH synthesis [21,22,23]. Two independent CEF pathways are described: the NAD(P)H dehydrogenase-like complex (NDH) dependent pathway and the Proton Gradient Regulation 5 (PGR5)-dependent pathway. The NDH consists of several subunits encoding NDH genes, and recent cryo-EM structural studies have provided molecular models for the electron transfer of the PSI-NDH complex [24,25]. However, NDH genes are missing from algae. Proton Gradient Regulation Like 1 (PGRL1) contributes to CEF, as well as switching between LEF and CEF in plants [16]. The PGRL1 protein contributes to CEF in *Chlamydomonas reinhardtii* [26,27,28,29] and is found in the *b*_6_*f* associated-PSI supercomplex [30]. In the vascular plant *Arabidopsis thaliana*, PGR5 was discovered to be an essential factor for CEF [31]. The PGR5 protein is conserved in algae and also contributes to CEF in *C*. *reinhardtii* [32]. PGR5 also enhances photosynthetic control and is important for low-potential chain redox tuning in *b*_6_*f*, particularly in CEF-activated conditions [33]. PGR5-dependent CEF is also important for avoiding PSII photoinhibition whether it is linked with NPQ or not [34].

Several articles describe the protection mechanisms of photoinhibition for both photosystems. FTSH contributes damaged-D1 degradation and also targets the *b*_6_*f* complex, suggesting that FTSH plays a major role in the quality control of thylakoid membrane proteins in the response of *C*. *reinhardtii* to light. As it is not yet clear whether FTSH targets PSI, we considered the effect of FTSH on PSI photoinhibition. To address this issue, we focused on PGR5, which is an essential factor for CEF and is conserved between plants and algae. We utilized the PGR5-disrupted mutant, *pgr5*, to facilitate the induction of PSI photoinhibition. We generated a *pgr5ftsh* double mutant and characterized its phenotype for PSI activity in response to high light treatment using the green algae *C. reinhardtii*.

## 2. Results

### 2.1. Growth Check

We crossed *pgr5* (mt^+^) and *ftsh1-1* (mt^−^) to generate the *pgr5ftsh* double mutant (Appendix A). All experiments included the *pgr5pgrl1* double mutant to consider the situation for abolishing CEF. To evaluate the qualitative cellular growth, we performed a spot test under continuous light (Figure 1). The cell suspensions were spotted on TAP or TP medium and exposed to different light intensities (0 to 200 μmol photons·m^−2^·s^−1^), with the LED having a 620 nm primary peak and a 453 nm secondary peak (Appendix A). The control and the PGR5-complemented strain (*pgr5* + PGR5) grew normally, indicating that the light source set up did not inhibit *C*. *reinhardtii* cellular growth. A previous report [32] showed that *pgr5* can grow photoautotrophically in up to 200 μmol photons·m^−2^·s^−1^ of continuous light, while its growth is significantly lower under fluctuating light conditions. However, we found that the photoautotrophic growth of *pgr5* (TP medium in Figure 1) was severely impaired even under 50 μmol photons·m^−2^·s^−1^ with our light source. These results indicate that our light source more effectively inhibited photosynthesis activity in *pgr5* than that in the previous report without fluctuating conditions.

### 2.2. Time Course of Fv/Fm

To roughly evaluate photoinhibition using this light source treatment, we monitored the time course of the Fv/Fm values after strong-light treatment. The Chlamydomonas cells were grown under low light (10 μmol photons·m^−2^·s^−1^), and then, exposed to high light (250 μmol photons·m^−2^·s^−1^) for 6 h, low light for 6 h, or high light for 2 h following low light for 5 h (Figure 2). The Fv/Fm values were decreased in all strains after 2 h of high-light incubation. Following an additional 4 h of high-light treatment, the Fv/Fm values remained 0.5 in the control, *pgr5pgrl1*, and *pgr5*-complemented strain (*pgr5* + PGR5). However, the Fv/Fm values decreased in *pgr5*, *ftsh1-1*, and *pgr5ftsh* with this trend being slightly milder in *pgr5*. To confirm the effect on photosystem protein accumulation during high-light treatment, we analyzed photosynthesis protein accumulation via immunoblotting (Appendix A). By adding chloramphenicol, which blocks chloroplast translation, we evaluated the stability of the chloroplast-encoded protein (Appendix A upper). We confirmed that the D1 protein level decreased in the presence of chloramphenicol during high-light exposure, and this effect became milder with an *ftsh1-1* background, which is consistent with previous reports [12,13]. However, we did not observe substantial changes in PSI reaction center protein accumulation. We observed an increment in the FTSH signal during high-light incubation, which is consistent with a previous report [35]. These results confirmed the effect of the *ftsh1-1* back-ground and that the accumulation levels of the PSI core protein was not affected under the high light condition. To investigate the recovery process after high-light treatment, we incubated the cells for up to 5 h under low light after 2 h of high-light treatment. The Fv/Fm values were slightly recovered in *pgr5*, *pgr5pgrl1*, *ftsh1-1*, and *pgr5ftsh*, while we confirmed complete recovery after 1 h of incubation in the control and *pgr5* + PGR5. These results indicate that our high-light treatment was able to induce photoinhibition, and following low-light treatment was able to recover the photosystem. Of note, Fv/Fm was decreased after 2 h of high-light incubation and remained around 0.5 in *pgr5pgrl1*, and *pgr5* followed this trend. This suggests complete CEF abolishment would potentially mitigate PSII photoinhibition. *pgr5ftsh* and *ftsh1-1* showed continuous Fv/Fm value decrement during high light, indicating that PSII was already photoinhibited, and the additional *pgr5* mutation did not rescue the phenotype for the Fv/Fm value in *pgr5ftsh*.

### 2.3. Time Course of Light-Induced Absorbance Changes at 705 nm

To evaluate the electron transfer activity of PSI, we measured light induced absorbance changes at 705 nm which correlate with P700 photooxidation [36,37,38]. The absorbance at 705 nm decreased in response to continuous actinic light illumination due to the accumulation of P700^+^. This absorbance decrement can be recovered through the cessation of actinic light illumination because of the reduction in P700^+^ resulting from electron transfer from plastocyanin, CEF, and back-reactions [38]. Because far-red light illumination barely induces an absorbance changes at 705 nm in *C*. *reinhardtii*, we illuminated the cells with red LEDs (represented as an orange ring, in the Materials and Method Section with a single peak at 630 nm). However, to evaluate light-induced PSI photo-oxidation activity, as 630 nm excites both PSI and PSII, we added DCMU to block PSII’s contribution, as well as DBMIB to eliminate the effect of electrons from the Cyt *b*_6_*f* complex on PSI [38]. DCMU and DBMIB were added to the sample just before taking the measurement. We observed that the 705 nm absorbance changes almost reached a plateau after 5 s of illumination; thus, we roughly estimated the effect on light-induced P700^+^ accumulation based on the amplitude just before the cessation of actinic light.

We found that the amplitude of absorbance changes during actinic light illumination was decreased in all strains after 2 h of high-light (250 μmol photons·m^−2^·s^−1^) treatment (Figure 3). Particularly, *pgr5pgrl1* showed no absorbance changes at 705 nm, and this did not change after 4 h, indicating that our light source setup effectively induced PSI photoinhibition without fluctuation. The *pgr5* strain also showed a strong reduction in the absorbance amplitude, which reduced further following 4 h of high-light treatment, indicating that it has a weaker phenotype compared with *pgr5pgrl1*. We barely observed any absorbance change at 705 nm after 6 h of high-light treatment. On the other hand, the control, *pgr5* + PGR5, *ftsh1-1*, and *pgr5ftsh* maintained their absorbance amplitudes even after 4 and 6 h of high-light treatment. This means that PSI function can be maintained and/or resist further PSI photoinhibition in *pgr5ftsh*, unlike the *pgr5* single mutant.

We also measured the light-induced absorbance changes in the cells during low-light (10 μmol photons·m^−2^·s^−1^) treatment after high-light (250 μmol photons·m^−2^·s^−1^) treatment (Figure 4). We reproduced the decrease in the light-induced absorbance change amplitude after 2 h of high-light treatment in all strains, and a particularly strong reduction was observed in *pgr5pgrl1* and *pgr5*. After 2 h of low-light treatment following 2 h of high-light treatment, the amplitude of the absorbance changes returned to the same level as before the high-light treatment in the control, *pgr5*+PGR5, and *pgr5ftsh*. We found that the amplitude was recovered, but the level was not the same as that before the high-light treatment in *pgr5* and *ftsh1-1*. After 4 h of low-light treatment, the amplitudes of *pgr5* and *ftsh1-1* returned to the same levels as before the high-light treatment. These results indicate that the PSI photoinhibition phenotype of *pgr5* was less observable in *pgr5ftsh*. However, the amplitude of *pgr5pgrl1* did not recover at all, even after 4 h of low light-incubation. That indicates that, in the case of the complete abolishment of CEF, our low-light treatment was not able to recover its PSI photoinhibition, and the 2-h high-light treatment inflicted irreversible damage on PSI.

## 3. Discussion

Under high-light treatment, in which CEF is activated, the electrons from PSI will be reinjected to the plastoquinone pool by CEF, and the electron-transfer rate within the *b*_6_*f* complex will be elevated through photosynthetic control. These mechanisms prevent an over-reduction in the plastoquinone pool under high light. An over-reduction in the PQ pool leads the formation of singlet oxygen, and then, plastoquinol scavenges singlet oxygen to protect D1 degradation [39,40,41]. Other PQ derivatives also work as potential ROS scavengers [42,43]. In the case of *ftsh1-1*, damaged PSII subunits, such as D1, were accumulated; additionally, we observed lower Fv/Fm during the high-light treatment, and the Fv/Fm-recovery rate became slower due to the lower rate of removal of damaged D1, which is consistent with a previous report (Figure 2) [12]. PSII deactivation did not affect P700 photo-oxidation kinetics when we poised the sample with DCMU and DBMIB, which was confirmed by the result of the light-induced absorbance changes at 705 nm; this possibility was also suggested in a previous report (Figure 3) [38]. The light-induced absorbance changes were not completely recovered after 2 h of low-light incubation in *ftsh1-1*, suggesting that damaged-PSII accumulation could also propagate PSI damage by reactive oxygen species (Figure 4). In the case of *pgr5*, PSI is photoinhibited, as confirmed in Figure 3, and the *b*_6_*f* complex electron-transfer rate is also inhibited according to Buchert, 2020 [33]. In this situation, the plastoquinone pool would be relatively more reduced compared to *ftsh1-1* but more oxidized compared to control; thus, the efficiency of ROS scavenging would be lower than in the control and less damaged-D1 accumulation would occur during high light. When we consider the low-light treatment after high-light treatment, the damaged-PSI-recovery rate is slower than in the control and is similar to that in *ftsh1-1* (Figure 4); this result in similar to the Fv/Fm-recovery rate for *pgr5* and *ftsh1-1* during low-light incubation after high-light treatment. These consideration are consistent with the trend of Fv/Fm decrement during high-light treatment (*pgr5* > *ftsh1-1*) and similar to the trend during low-light treatment after high-light treatment (Figure 2).

When we completely abolish CEF, such as in *pgr5pgrl1*, the plastoquinone pool is more oxidized than in *pgr5*, while the electron-transfer rate of the *b*_6_*f* complex is also slowed down. As shown in Figure 3 and Figure 4, the light-induced absorbance change was completely lost after 2 h of high-light treatment and did not recover, even after 4 h of low-light treatment. Therefore, the plastoquinone redox state is more reduced compared to *pgr5* after high-light treatment, which would enable it to more effectively scavenge single oxygen to protect D1 degradation. During low-light treatment after high light-treatment, electron transfer activity will not recover completely or will recover at the slowest rate because PSI activity is already impaired with high-light treatment. These considerations are consistent with our observation of Fv/Fm under high-light treatment and following low-light incubation, in which the reduced Fv/Fm value was maintained during high-light treatment and the slowest Fv/Fm-recovery rate occurred during low-light incubation (Figure 2). From another perspective, the complete abolishment of CEF would not generate less ΔpH to drive the synthesis of ATP molecules, therefore, its repair efficiency would be slowed down, resulting in the slowest Fv/Fm-recovery rate during low light after high-light treatment.

In the double-mutant *pgr5ftsh*, we observed that the time course of Fv/Fm during high light and during low light after high light exhibited the same trend as *ftsh1-1*, and the *pgr5* phenotype was masked (Figure 2). This indicates the *pgr5ftsh* double mutant accumulates damaged PSII in the same way as the *ftsh1-1* single mutant, as confirmed via immunoblotting (Appendix A). The light-induced P700 photo-oxidation kinetics also exhibited the same trend with *ftsh1-1* during both high light and low light after high light (Figure 3 and Figure 4). The *pgr5* mutation decreases electron reinjection into the plastoquinone pool, as well as the *b*_6_*f* complex electron-transfer rate; thus, PSI photoinhibition should occur as observed in the single mutant (Figure 3 and Figure 4). In the case of the *pgr5ftsh* double mutant, damaged-PSII accumulation would enable less electron transfer from PSII to the plastoquinone pool, which could lead to the plastoquinone pool exhibiting less over-reduction than *pgr5* (because *pgr5* does not accumulate damaged PSII). Lower over-reduction of the plastoquinone pool will generate less singlet oxygen [39], which would cause less damage to PSI. In addition, the electron-transfer rate from plastoquinone pool to PSI would be slowed down due to lowered *b*_6_*f* electron-transfer rate in *pgr5*’s genetic background. Therefore, mitigated PSI photoinhibition was observed (Figure 5). Another possibility is that FTSH is also involved in damaged-PSI subunit removal. In this scenario, oxidized PSI subunits (which might be enhanced by the *pgr5* mutation) would be recognized by FTSH and removed. Unlike PSII, oxidized PSI subunits might be more stable because they have been constitutively detected in field-grown spinach. In addition, slower PSI-synthesis rate and the removal of PSI subunits would lead to irreversible/hard-to-recover PSI photoinhibition. However, intensive biochemical experiments are required to show that FTSH targets PSI as a substrate, which is impossible to confirm from the present data. The limitation of PSI photoinhibition in *pgr5ftsh* was found to be an experimentally observable phenotype within a period of several hours period. In fact, the cellular growth of *pgr5ftsh* was impaired in the photoautotrophic condition and strongly reduced in the photomixotrophic condition under the higher-light condition (more than 100 μmol photons·m^−2^·s^−1^) (Figure 1).

## 4. Materials and Methods

### 4.1. Strains, Growth, Crossing, Spot Test and Immunoblotting

The *Chlamydomonas reinhardtii* cells were grown in Tris-acetate phosphate (TAP), Tris-phosphate (TP) medium, or minimum medium (MM) under low (10 μmol photons·m^−2^·s^−1^) or high light intensity (250 μmol photons m^−2^ s^−1^). Please refer to the Section 4.2 for the equipment setup. The *pgr5* (mt^+^) [32] and *ftsh1-1* (mt^−^) strains [12] were crossed according to Harris [44]. To determine *ftsh1-1*’s genetic background, the 333 bp DNA fragment was prepared via PCR with L1 and R2 primers, including the *fts1-1* mutation site which is not digested, using EagI resriction enzyme in *ftsh1-1* (Appendix A). The PCR product was digested with EagI and 207 bp and 126 bp fragments were generated from the wild-type sequence of *FTSH1*’s genetic background, not from *ftsh1-1*’s background (Appendix A). We determined *pgr5*’s genetic background in the photomixotrophic growth condition under ambient light supplemented with 20 μg paromomycin·mL^−1^ as well as genotyping with two independent PCRs using L2 + R2 and L3 + R3, of which the genomic DNA was deleted and replaced with an antibiotic resistance marker cassette in *pgr5*’s background. Finally, we obtained the *pgr5ftsh* double mutant (Appendix A). The *pgr5*-complemented strain, *pgr5* + PGR5, was from Johnson et al. [32]. The *pgr5pgrl1* double mutant was from Steinbeck et al. [45]. Cells grown in liquid TAP medium under low light were suspended in TP medium and incubated for 12 h under low light. To perform the high-light treatment, the cell concentration was adjusted with fresh TP medium at 20 μg chlorophyll·mL^−1^ and incubated for several hours with shaking under high light. To maintain uniform brightness on the surface of the cell culture, cell suspensions were placed in a low-form beaker, and the depth of the cell suspension in the beaker was no more than 5 mm. To perform spot test, the cell concentration was adjusted with fresh TP medium at 25 ng chlorophyll·μL^−1^, and then, 4 μL of suspension was spotted on a TAP or TP medium plate. SDS-PAGE followed by immunoblotting for whole-cell polypeptides were performed according to a previous article [46]. The signals from enhanced chemiluminescence were detected using ChemiDoc (BioRad, Hercules, CA, USA).

### 4.2. Light Source

For the measurement and spot test for the high-light treatment and the following low-light treatment, pink and/or yellow PLANTFLECs (NK system, Osaka, Japan) were used in the following combinations. For the high-light treatment and spot test, three pink PLANTFLECs and three yellow PLANTFLECs were placed alternately, and a 620 nm primary emission peak and a 453 nm secondary emission peak were prepared (Appendix A). The light intensities for the spot test were adjusted by changing the distance between the light source and plates or by applying several sheets of white papers. For the following low-light treatment, a yellow PLANTFLEC was used (according to the instrument manual, the yellow PLANTFLEC has a 600 nm primary emission peak and a 453 nm secondary emission peak). The emission spectrum of the light source was recorded with a light analyzer LA-105 (NK system, Japan).

### 4.3. Chlorophyll Fluorescence Measurement and Absorbance Changes in Cells at 705 nm

Cells were suspended with liquid TP medium at 20 μg chlorophyhll·mL^−1^ before measurement. Chlorophyll fluorescence measurement was performed with a DUAL-PAM-100 (Walz, Effeltrich, Germany). For the measurement of light-induced absorbance changes at 705 nm, the cell suspension was mixed with the same volume of TP medium containing 20% (*w*/*v*) Ficol, and DCMU (1 μM) and DBMIB (1 μM) were added just before measurement. The samples were placed for 30 s in the darkness in a sample cuvette holder and illuminated with an orange actinic light at an intensity of 82 μmol photons m^−2^ s^−1^ for 5 s, followed by darkness for 5 s. The absorbance changes were recorded with a JTS-10 (BioLogic, Seyssinet-Pariset, France) using the following sequence program: 3(10msD)5s2(10ms)10msI{200µs,35,5s,J200µsD15µsI}20µsJ200µsD{2ms,35,5s,D}50msD.

### 4.4. PCR and the Following Restriction Enzyme Digestion

The total DNA of *C*. *reinhardtii* was extracted from cells suspended in 5% (*w*/*v*) Chelex (BioRad, Hercules, CA, USA) by heating them at 100 °C for 10 min. For *ftsh1-1* genotyping, L1 (5′-TCCAAGGCGCCCTGCATCATC-3′) and R1 (5′-GCGGCGCGCGATCTTCTCCAG-3′) primers were used to generate the PCR product, which was diluted two times and incubated for 1–2 h at 37 °C after adding EagI (NEB, Ipswich, MA, USA) to distinguish the *ftsh1-1* mutation. For *pgr5* genotyping, L2 (5′-CAGCGTAAAGCACGGTATCA-3′), L3 (5′-CTCGCAGCCAAAACACATTA-3′), R2 (5′-GTAGCCTTGTTGCCCATCAT-3′), and R3 (5′-GGGTAAAAAGCCATGTCAGG-3′) primers were used to amplify the deleted region in the *pgr5* mutant.

## Figures and Tables

**Figure 1 plants-13-00606-f001:**
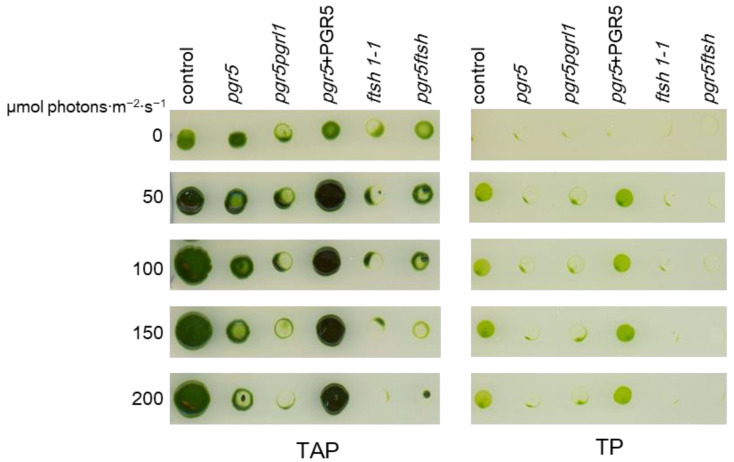
Spot test on solid medium. Corresponding 100 ng chlorophyll cell suspensions were spotted on TAP or TP medium and incubated under different light intensities emitted from the light source shown in Appendix A. A representative solid plate is shown here. Cells prepared from the third biologically replicated batch were spotted on solid medium and incubated for 5 days at 25 °C under each light intensity.

**Figure 2 plants-13-00606-f002:**
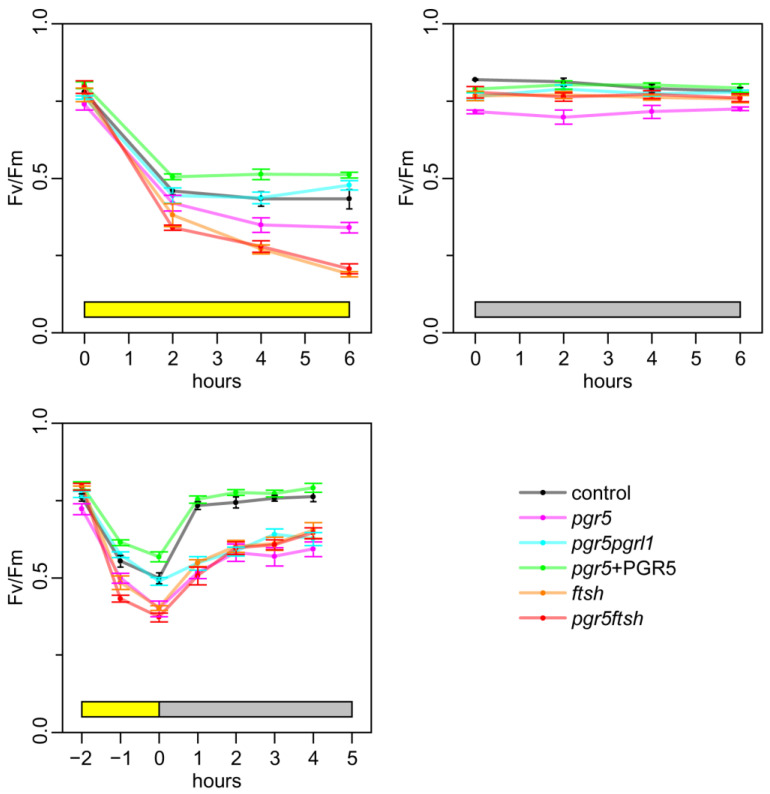
Time course of Fv/Fm under light treatment. Cells were incubated under high light for 6 h (**upper left**) and low light for 6 h (**upper right**), and underwent 2 h of high-light treatment following 5 h of low-light treatment (**lower left**). The Fv/Fm value was evaluated every 2 h in the only high-light and only low-light treatment, and every hour in the high-light and treatment followed by recovery. The mean values with standard error bars from three biological replicates were plotted, and each point was connected with a straight line. The color codes are as follows: control (gray), *pgr5* (magenta), *pgr5pgrl1* (cyan), *pgr5* + PGR5 (green), *ftsh1-1* (yellow), and *pgr5ftsh* (red).

**Figure 3 plants-13-00606-f003:**
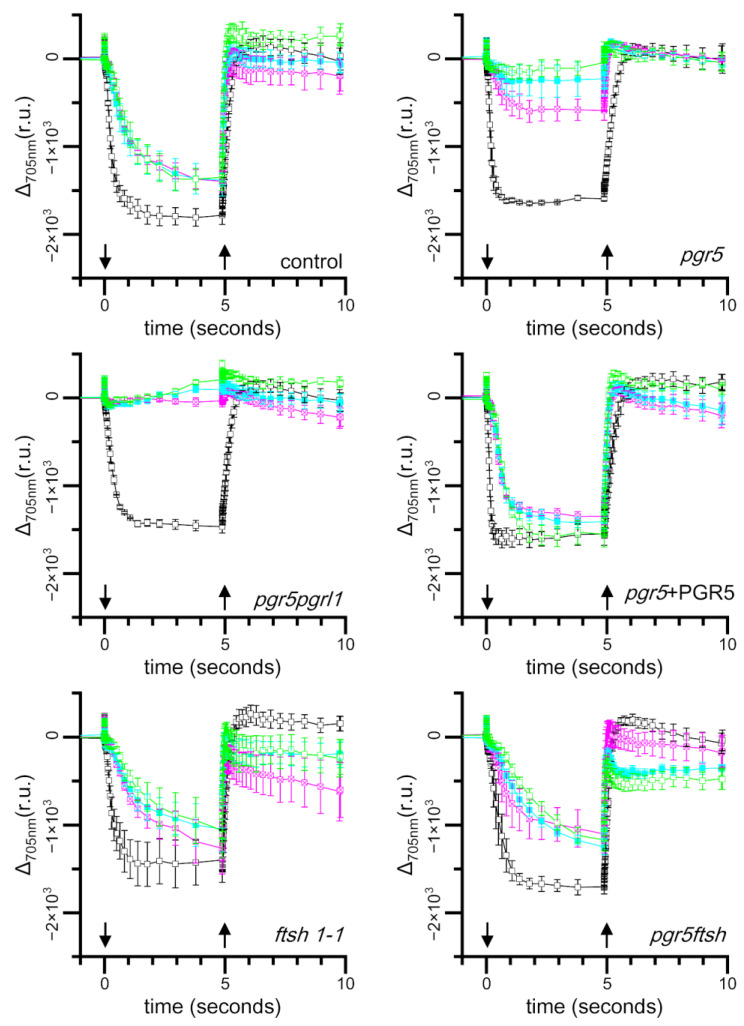
Light-induced absorbance changes at 705 nm from the cells incubated under high light. Cells were sampled from the cell culture incubated under high light every 2 h, and the light-induced absorbance changes at 705 nm were measured. The color codes of the traces are as follows: before high light (black); 2 h (magenta), 4 h (cyan), and 6 h (green) after high-light treatment. The strain names are shown at the bottom right in each panel. The cells were poised with 1 μM DCMU and 1 μM DBMIB just before measurement. The actinic light was switched on and off at points indicated by downward and upward arrows, respectively. The mean values with standard error bars from three biological replicates were plotted and connected with straight lines.

**Figure 4 plants-13-00606-f004:**
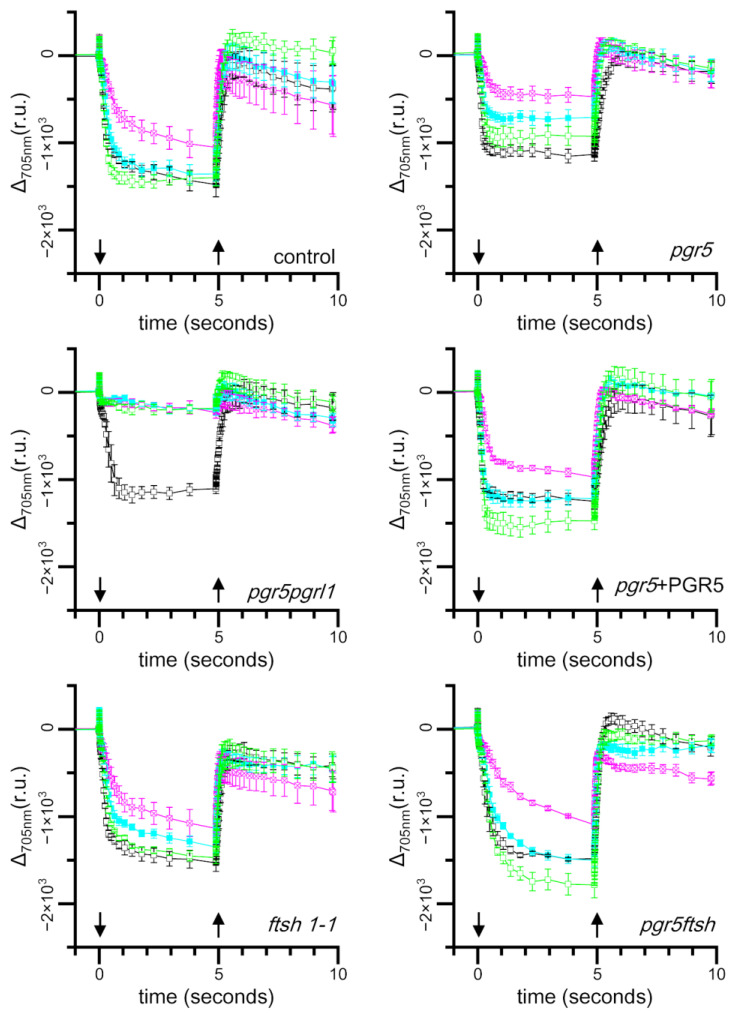
Light-induced absorbance changes at 705 nm from the cells incubated under high light and followed by low light. Cells was sampled from the cell culture incubated under high light followed by low light every 2 h, and the light-induced absorbance changes at 705 nm were measured. The color codes of the traces are as follows: before high-light treatment (black); 2 h after high light (magenta); and 2 h (cyan) and 4 h (green) after low-light treatment. The strain names are shown at the bottom right in each panel. The cells were poised with 1 μM DCMU and 1 μM DBMIB just before measurement. The actinic light was switched on and off at points indicated by downward and upward arrows, respectively. The mean values with standard error bars from three biological replicates were plotted and connected with straight lines.

**Figure 5 plants-13-00606-f005:**
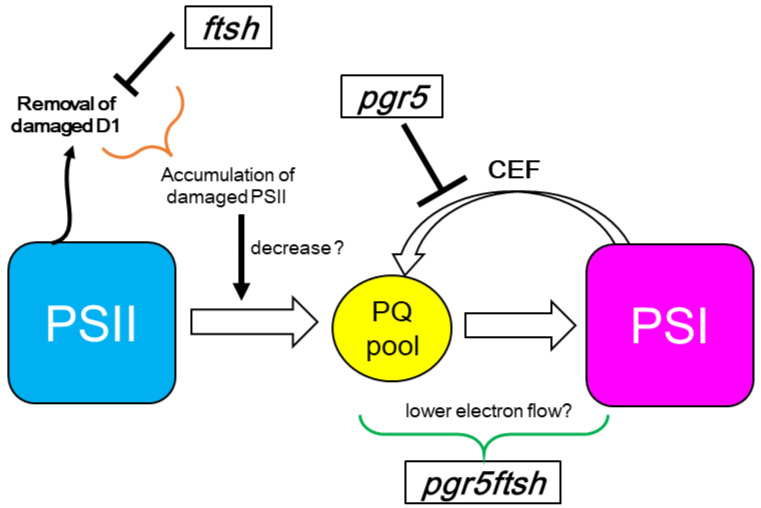
Possible hypothetical model for electron flow in *pgr5ftsh* double mutant. The *pgr5* mutation blocks CEF, resulting in lower electron reinjection to the plastoquinone pool (PQ pool) and lowered *b*_6_*f* electron transfer rate. The *ftsh* mutation leads to the accumulation of damaged PSII and additional mutation to *pgr5* mutation background results in relatively less overreduction of the PQ pool. This situation leads lower electron transfer from PSII to PQ pool. A mitigated PSI photoinhibition phenotype was observed in the *pgr5ftsh* mutant.

## Data Availability

Data are contained within the article and Appendix A.

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
