# Peer review of "Dysfunction of Chloroplast Protease Activity Mitigates pgr5 Phenotype in the Green Algae Chlamydomonas reinhardtii"

_plants, 2024, doi:10.3390/plants13050606_

Round 1
Reviewer 1 Report
Comments and Suggestions for Authors
Paper Ozawa et al. described the functional characterization of pgr5 phenotype in green algae C. reinhardtii, story is interesting for the photosynthetic research field, and although, the overall presentation and writing of the manuscript is good, I would suggest moderate revision and paper need careful editing.
Introduction/discussion sections are missing some important references, it should include recent updates in field specially about PSII photoinhibition process which leads to ROS formation. Furthermore, the reduction of PQ (plastoquinol), and PQ-derivatives which are known antioxidant to mitigate the PSII damage.
In discussion, line 239-242… Author described the prevention mechanisms for overreduction of PQ-pool under photo-inhibitory conditions, but completely ignored the overreduction of PQ pool which leads the formation of singlet oxygen (1O2), (Vass, 2012 Biochim Biophys Acta, 1817(1):209-217, and others) and subsequently reduced PQ (plastoquinol) scavenge singlet oxygen to protect D1 degradation (Kruk and Trebst 2008, Biochim Biophys Acta 1777(2) 154-162; Yadav et al. 2010, Biochim Biophys Acta 1797(11) 1807-1811 and others), not only this but also other PQ-derivatives works as potential ROS scavenger such as PC (Rastogi et al. 2014, Plant cell and Environment 37, 3920401; Nowicka etal. 2021, Int J Mol Sci 22(6): 2950 and others), the mechanism for the formation and scavenging of ROS due to overreduction of PQ-pool under photo-inhibitory condition should also be considered and discussed.
figure 5; As hypothetical model needs to be confirmed with additional studies, therefore should have visible questions mark (left or right side), so readers should not take it conclusive, because these days quick google image search would show such model (image) and if student/reader does not read full article, chances are that it would be misinterpreted.
paragraph from results (section 2.1), about the generation of double mutant would be better to move in methods and materials.
Author Response
>Paper Ozawa et al. described the functional characterization of pgr5 phenotype in green algae C. reinhardtii, story is interesting for the photosynthetic research field, and although, the overall presentation and writing of the manuscript is good, I would suggest moderate revision and paper need careful editing.
Thank you for your comments. We revised manuscript.
>Introduction/discussion sections are missing some important references, it should include recent updates in field specially about PSII photoinhibition process which leads to ROS formation. Furthermore, the reduction of PQ (plastoquinol), and PQ-derivatives which are known antioxidant to mitigate the PSII damage.
>In discussion, line 239-242… Author described the prevention mechanisms for overreduction of PQ-pool under photo-inhibitory conditions, but completely ignored the overreduction of PQ pool which leads the formation of singlet oxygen (1O2), (Vass, 2012 Biochim Biophys Acta, 1817(1):209-217, and others) and subsequently reduced PQ (plastoquinol) scavenge singlet oxygen to protect D1 degradation (Kruk and Trebst 2008, Biochim Biophys Acta 1777(2) 154-162; Yadav et al. 2010, Biochim Biophys Acta 1797(11) 1807-1811 and others), not only this but also other PQ-derivatives works as potential ROS scavenger such as PC (Rastogi et al. 2014, Plant cell and Environment 37, 3920401; Nowicka etal. 2021, Int J Mol Sci 22(6): 2950 and others), the mechanism for the formation and scavenging of ROS due to overreduction of PQ-pool under photo-inhibitory condition should also be considered and discussed.
Thank you for your comments. We revised the discussion section according to your suggestions.
>figure 5; As hypothetical model needs to be confirmed with additional studies, therefore should have visible questions mark (left or right side), so readers should not take it conclusive, because these days quick google image search would show such model (image) and if student/reader does not read full article, chances are that it would be misinterpreted.
Thank you for your comments. We revised figures.
>paragraph from results (section 2.1), about the generation of double mutant would be better to move in methods and materials.
We revised the result. We moved double mutant generation procedure to the materials and methods.
Reviewer 2 Report
Comments and Suggestions for Authors
In this manuscript the authors have dissected the contribution of two mechanisms involved in PSII and PSI photoprotection in response to high-light treatments, by exploiting Chlamydomonas reinhardtii strains mutated in cyclic electron flow protein Pgr5 and in metalloprotease FtsH1.
Results are rather interesting and deserve publication in “plants”. However, few concerns need to be addressed:
In plants CEF is required for adaptation to fluctuating light and pgr mutants result lethal in such condition. Is the same in Chlamydomonas?
What’s the contribution of Pgr5 and FtsH1 in response to fluctuating light? To assess this, authors are invited to test pgr5, ftsh1 and pgr5ftsh1 mutants in changing light conditions by evaluating PSII and PSI parameters.
Figure 1: TP growth is rather weak and it’s difficult to properly determine growth. I suggest to repeat growth experiments to better state differences, including growth-rate quantification (in liquid medium?) and by allowing cells to grow more in spot test on TP medium (Figure 1).
How many plastid-located FtsH proteins are encoded in Chlamydomonas genome? Is FtsH1 the main one involved in PSII repair?
Furthermore, is the ftsh1-1 mutant a loss of function? Please clarify and discuss that.
Figure 3, Line 138-140: “Black, magenta, cyan, and green color traces represent before high light treatment, 2, 4, 6 hours after high light treatment, respectively ”. This is confusing, please rephrase the sentence.
Comments on the Quality of English Language
Results are rather interesting and deserve publication in “plants”. However, the writing is rather sloppy and there are grammar mistakes, the manuscript requires extensive English language editing to be considered for publication.
Author Response
>In this manuscript the authors have dissected the contribution of two mechanisms involved in PSII and PSI photoprotection in response to high-light treatments, by exploiting Chlamydomonas reinhardtii strains mutated in cyclic electron flow protein Pgr5 and in metalloprotease FtsH1.
>Results are rather interesting and deserve publication in “plants”. However, few concerns need to be addressed
Thank you for your comments. We revised manuscript.
>In plants CEF is required for adaptation to fluctuating light and pgr mutants result lethal in such condition. Is the same in Chlamydomonas?
The case is similar with plants. The Chlamydomonas pgr5 mutant survived, but it had a significantly lower growth rate under the mild fluctuating light condition (20 μmol photons m-2 s-1 for 5 minutes/200 μmol photons m-2 s-1 for 30 seconds) and was unable to grow under the severe fluctuating light condition (20 μmol photons m-2 s-1 for 5 minutes/600 μmol photons m-2 s-1 for 30 seconds). The Chlamydomonas pgrl1 mutant showed impaired growth under the severe fluctuating light condition but not in the mild fluctuating light condition. However, neither mutant displayed a significant growth rate defect under continuous light, even at 600 μmol photons m-2 s-1. These phenotypes are reported in (Jokel, M. et al. Plant J 94, 822-835 (2018).). We mentioned these phenotypes briefly in the Introduction Section in the revised manuscript.
>What’s the contribution of Pgr5 and FtsH1 in response to fluctuating light? To assess this, authors are invited to test pgr5, ftsh1 and pgr5ftsh1 mutants in changing light conditions by evaluating PSII and PSI parameters.
PGR5 contributes to PSI photoprotection under fluctuating light stress, but it is not yet clear whether this is the case for FTSH1. Unlike plants, Chlamydomonas (as well as mosses) contains FLV, PGR5, and PGRL1, which are involved in fluctuating light photoprotection. We have to include FLV mutants and their complemented lines in our dataset if we consider fluctuating light, which should be prepared in another independent article. The purpose of the high-light treatment is to induce photoinhibition. We observed photoinhibition in both photosystems in the present experiment.
>Figure 1: TP growth is rather weak and it’s difficult to properly determine growth. I suggest to repeat growth experiments to better state differences, including growth-rate quantification (in liquid medium?) and by allowing cells to grow more in spot test on TP medium (Figure 1).
The purpose of the spot test is to show a qualitative effect on cellular growth using the same light source as for the high-light treatment. We mentioned this in the revised manuscript. The control and PGR5-complemented strains were grown in TP medium. As 10 μmol photons m-2 s-1 is technically difficult to control under experimental conditions, we removed the 10 μmol photons m-2 s-1 result in the revised manuscript and decided not to discuss the physiological effect at this light intensity.
>How many plastid-located FtsH proteins are encoded in Chlamydomonas genome? Is FtsH1 the main one involved in PSII repair?
>Furthermore, is the ftsh1-1 mutant a loss of function? Please clarify and discuss that.
Two FTSH proteins (FTSH1 and FTSH2) are localized in chloroplasts in Chlamydomonas reinhardtii and enable the hetero hexamer to function. The ftsh1-1 strain is a loss-of-function mutant, and previous work showed that FTSH1 is involved in PSII repair (Plant Cell 26, 373-390 (2014)). We included this information in the Introduction Section of the revised manuscript.
>Figure 3, Line 138-140: “Black, magenta, cyan, and green color traces represent before high light treatment, 2, 4, 6 hours after high light treatment, respectively ”. This is confusing, please rephrase the sentence.
We rephrased the following from Figure 4: “The color codes of traces are following: before high light (Black), 2 hours (magenta), 4 hours (cyan), and 6 hours (green) after high light treatment.” to “The color codes of the traces are as follows: before high-light treatment (black); 2 hours after high light (magenta); and 2 hours (cyan) and 4 hours after low-light treatment.”
Reviewer 3 Report
Comments and Suggestions for Authors
In the manuscript, Ozawa et al. generated a pgr5ftsh double mutant in Chlamydomonas reinhardtii and analyzed the activity of PSI and PSII compared to the single mutants and a pgr5pgrl1 double mutant under different light conditions. The data indicate that the loss of PGR5 and FTSH leads to reduced photoinhibition of PSI compared to the pgr5 mutant. Overall, the data support the conclusions. However, the manuscript could be improved in the following points:
- Fig. S1: The absence of PGR5 in the pgr5ftsh double mutant should be verified by genotyping or Western blot analysis. In my opinion, it is not sufficient to verify the genetic background only by testing growth on paromomycin-containing medium. Furthermore, it should be explained in the text why the R420C mutation in FTSH leads to a loss of FTSH function.
- Fig. 1: The intention to include the pgr5pgrl1 double mutant in the experiments should be better explained in the main text. The authors should also discuss why this double mutant shows better growth than pgr5 at low light.
- Fig.S3: The western blot data to show degradation of PSAD/F and CP1 after 8 h in pgr5 are not convincing. The differences are very subtle and could be due to small differences in loading. In my opinion, the manuscript would not loose quality if these data were not presented.
- Fig.5: The title is mentioned twice. The legend does not contain any explanations of the model. This should be added. Otherwise, the model is not understandable.
- The abstract should better summarize which results support the conclusion of the last sentence “We concluded…”. In particular, the sentence "We found light induced P700 photooxidation kinetics was similar trend with ftsh1-1 and control." is not clear.
Comments on the Quality of English LanguageThe English language needs to be corrected throughout the text.
Author Response
>In the manuscript, Ozawa et al. generated a pgr5ftsh double mutant in Chlamydomonas reinhardtii and analyzed the activity of PSI and PSII compared to the single mutants and a pgr5pgrl1 double mutant under different light conditions. The data indicate that the loss of PGR5 and FTSH leads to reduced photoinhibition of PSI compared to the pgr5 mutant. Overall, the data support the conclusions. However, the manuscript could be improved in the following points:
Thank you for your comments. We revised manuscript.
>- Fig. S1: The absence of PGR5 in the pgr5ftsh double mutant should be verified by genotyping or Western blot analysis. In my opinion, it is not sufficient to verify the genetic background only by testing growth on paromomycin-containing medium. Furthermore, it should be explained in the text why the R420C mutation in FTSH leads to a loss of FTSH function.
Thank you for your comments. We added data for the genotyping of the PGR5 gene for the pgr5ftsh double mutant in Figure S1. FTSH1-R420C fails to form a functional hetero hexamer, as reported in (Plant Cell 26, 373-390 (2014)). We added this description to the Introduction Section.
>- Fig. 1: The intention to include the pgr5pgrl1 double mutant in the experiments should be better explained in the main text. The authors should also discuss why this double mutant shows better growth than pgr5 at low light.
Thank you for your comments. The reason to include the pgr5pgrl1 is for the complete abolishment of CEF. We revised manuscript. The purpose of this spot test experiment is to show qualitative cellular growth using the same light source for high-light treatment. As 10 μmol photons m-2 s-1 is technically difficult to control, we removed the 10 μmol photons m-2 s-1 result in the revised manuscript and decided not to discuss the physiological effect at this low light intensity.
>- Fig.S3: The western blot data to show degradation of PSAD/F and CP1 after 8 h in pgr5 are not convincing. The differences are very subtle and could be due to small differences in loading. In my opinion, the manuscript would not loose quality if these data were not presented.
Thank you for your comment. We removed the PSAD/F data in the revised manuscript. We kept the CP1 data to highlight that the PSI core protein accumulation level was not changed substantially.
>- Fig.5: The title is mentioned twice. The legend does not contain any explanations of the model. This should be added. Otherwise, the model is not understandable.
Thank you for your comment. We revised the figure and added legends for clarification.
>- The abstract should better summarize which results support the conclusion of the last sentence “We concluded…”. In particular, the sentence "We found light induced P700 photooxidation kinetics was similar trend with ftsh1-1 and control." is not clear.
Thank you for your comments. We revised the Abstract to provide a more precise description of the experiment results.
Reviewer 4 Report
Comments and Suggestions for Authors
The manuscript by Shin-Ichiro Ozawa, Zhang Guoxian, and Wataru Sakamoto entitled ‘Dysfunction of chloroplast protease activity mitigates pgr5 phe-2 notype in the green algae Chlamydomonas reinhardtii’ was reviewed. In my opinion, the manuscript is very confusingly written, the text needs proofreading, and some conclusions are questionable. I think that the manuscript can not be published in the current view. My main comments are listed below.
In the beginning of the Introduction (L27), the authors try to describe the electron-transport-chain of the thylakoid membrane. I would recommend to the authors more detailed read any review about this because many mistakes are present here. For example, charge separation occurs in PSII or PSI (between their donor and acceptor sides), but I don’t know what is ‘charge separation of special pair chlorophylls’. P680 and P700 can only become oxidized.
‘The sum of electro potential (Δψ) by electron transfer and proton gradient across thylakoid membrane (ΔpH) are known to be proton motive force (pmf)’ (L53) is not correct. Electro potential (Δψ) is formed by ions transferred across the thylakoid membrane, but not by electrons.
L57 ‘Under strong light, in which both photosystems are potentially overexcited and to be protected from photoinhibition by different strategy: protein regeneration for PSII and electron flow regulation for PSI’ is not correct. PSII needs to be more protected from HL, because it is a most sensitive complex. The NPQ is one of them. The protein regeneration is a consequence of the PSII photodegradation already.
L67 ‘It is reported that oxidation of the stromal side D1 subunit is important for the interaction with FTSH [4] and several oxidation sites are found in PSI subunit’. What is a subunit?
L73 ‘to generate sufficient amount of pmf to establish NPQ’. NPQ is induced by ΔpH only.
L 93 ‘how these two mechanisms are involved’. Why do the authors only talk about two photoprotection mechanisms? In addition, I disagree that D1 replacement is a mechanism of photoprotection. In contrast, the main part of PSII photoprotection comes from NPQ mechanisms, including qE (PsbS), qZ(VDE-cycle), state transitions, in addition by change in antenna size. What about this wide front of the photoprotection mechanisms? Why the do authors not take them into account in the study?
L138. This was difficult to find how many units HL and LL are.
600 mkmol photons is too strong for Chlamydomonas. The authors should also study the light intensity near 300 mkmol photons, which are usually used as HL for Chlamydomonas.
The Western blot (S3) shows very interesting results, but the lack of repeats makes them questionable.
L242. The prevention of the PQ-pool over-reduction is realized mainly by ‘closing’ of PSII, but not by the PSII degradation.
and etc.
The authors present a scheme (Fig. 5), but it is not clear where in it the different NPQ mechanisms should be located, because their involvement in PSII photoprotection is quite certain.
Thus, I think that the authors should completely rewrite their manuscript, make the idea clearer, describe the results more accurately, and take into account other photoprotection mechanisms.
Comments on the Quality of English Languageshould be corrected
Author Response
>The manuscript by Shin-Ichiro Ozawa, Zhang Guoxian, and Wataru Sakamoto entitled ‘Dysfunction of chloroplast protease activity mitigates pgr5 phenotype in the green algae Chlamydomonas reinhardtii’ was reviewed. In my opinion, the manuscript is very confusingly written, the text needs proofreading, and some conclusions are questionable. I think that the manuscript can not be published in the current view. My main comments are listed below.
Thank you for your comments. We rewrote the manuscript. The revised manuscript focuses on the effect of additional ftsh mutations in the CEF mutant.
>In the beginning of the Introduction (L27), the authors try to describe the electron-transport-chain of the thylakoid membrane. I would recommend to the authors more detailed read any review about this because many mistakes are present here. For example, charge separation occurs in PSII or PSI (between their donor and acceptor sides), but I don’t know what is ‘charge separation of special pair chlorophylls’. P680 and P700 can only become oxidized.
>‘The sum of electro potential (Δψ) by electron transfer and proton gradient across thylakoid membrane (ΔpH) are known to be proton motive force (pmf)’ (L53) is not correct. Electro potential (Δψ) is formed by ions transferred across the thylakoid membrane, but not by electrons.
>L57 ‘Under strong light, in which both photosystems are potentially overexcited and to be protected from photoinhibition by different strategy: protein regeneration for PSII and electron flow regulation for PSI’ is not correct. PSII needs to be more protected from HL, because it is a most sensitive complex. The NPQ is one of them. The protein regeneration is a consequence of the PSII photodegradation already.
Thank you for your comments. We revised the manuscript according to your suggestions.
>L67 ‘It is reported that oxidation of the stromal side D1 subunit is important for the interaction with FTSH [4] and several oxidation sites are found in PSI subunit’. What is a subunit?
We added information on the subunits described in the reference. The reference reported oxidation sites in the PsaA/B/C/D/E/F/H/J/L and LHCI subunits (Lhca1/2/3/4).
>L73 ‘to generate sufficient amount of pmf to establish NPQ’. NPQ is induced by ΔpH only.
Thank you for your comment. We corrected this in the revised manuscript.
>L 93 ‘how these two mechanisms are involved’. Why do the authors only talk about two photoprotection mechanisms? In addition, I disagree that D1 replacement is a mechanism of photoprotection. In contrast, the main part of PSII photoprotection comes from NPQ mechanisms, including qE (PsbS), qZ(VDE-cycle), state transitions, in addition by change in antenna size. What about this wide front of the photoprotection mechanisms? Why the do authors not take them into account in the study?
Thank you for your comments. We simplified the text and considered the effect of the ftsh mutation on PSI photoinhibition. We also mentioned PSII protection mechanisms in the revised Introduction Section.
>L138. This was difficult to find how many units HL and LL are.
We added light intensity information for each experiment.
>600 μmol photons is too strong for Chlamydomonas. The authors should also study the light intensity near 300 μmol photons, which are usually used as HL for Chlamydomonas.
Thank you for your comments. We reconfirmed that the light source instrument emits around 250 and no more than 300 μmol photons, not 600 μmol photons. We corrected the value in the revised manuscript.
>The Western blot (S3) shows very interesting results, but the lack of repeats makes them questionable.
We took out the dataset for the PSAD/F LHCA2 subunits from the Western blot analysis because the differences in signal intensity are small and less quantitative for these subunits. However, the results for D1 and FTSH are consistent with previous reports and support our light treatment experiment.
>L242. The prevention of the PQ-pool over-reduction is realized mainly by ‘closing’ of PSII, but not by the PSII degradation.
Thank you for your comments. We removed this sentence.
>The authors present a scheme (Fig. 5), but it is not clear where in it the different NPQ mechanisms should be located, because their involvement in PSII photoprotection is quite certain.
We revised Figure 5. To simplify the scheme, we indicated the damaged D1 which is involved in the ftsh mutation and added CEF to more precisely highlight the pgr5 mutation.
>Thus, I think that the authors should completely rewrite their manuscript, make the idea clearer, describe the results more accurately, and take into account other photoprotection mechanisms.
Thank you for your suggestions. We described other photoprotection mechanisms for PSII in the Introduction. We thoroughly reorganized the manuscript to more accurately describe the results and make the idea clearer. We hypothesized that FTSH also targets PSI as a substrate, and thus, we focused on PSI photoinhibition and observed the effect by adding the FTSH loss-of-function mutation to the pgr5 mutant. Unfortunately, biochemical works are not sufficient to address this issue, but for the moment, we can report that the pgr5 mutation was masked by an additional ftsh mutation based on a light-induced absorbance change.
Round 2
Reviewer 4 Report
Comments and Suggestions for Authors
The manuscript by Shin-Ichiro Ozawa, Zhang Guoxian, and Wataru Sakamoto entitled ‘Dysfunction of chloroplast protease activity mitigates pgr5 phenotype in the green algae Chlamydomonas reinhardtii’ was reviewed again after the authors revised it significantly.
This is obviously, that a grate work was made with the text, and now I don’t have any serious questions about the manuscript.
I only should indicate to the authors that the beginning of the Introduction (from L34 to 84) obviously requires insert of relevant citations. After that correction the manuscript will be able to published.
Author Response
>The manuscript by Shin-Ichiro Ozawa, Zhang Guoxian, and Wataru Sakamoto entitled ‘Dysfunction of chloroplast protease activity mitigates pgr5 phenotype in the green algae Chlamydomonas reinhardtii’ was reviewed again after the authors revised it significantly.
>This is obviously, that a grate work was made with the text, and now I don’t have any serious questions about the manuscript.
>I only should indicate to the authors that the beginning of the Introduction (from L34 to 84) obviously requires insert of relevant citations. After that correction the manuscript will be able to published.
Thank you for your comment.
We inserted citations at the beginning of the introduction.
Round 3
Reviewer 4 Report
Comments and Suggestions for Authors
I thank the authors for the work done and for improving the quality of the manuscript. Thus, I think that the manuscript can be accepted for publication in its current form.